# Offline events and online hate

Yonatan Lupu[1,2]*, Richard Sear[3], Nicolas Velásquez[4], Rhys Leahy[1,2], Nicholas Johnson Restrepo[2], Beth Goldberg[5], Neil F. Johnson[2,3]

**1** Political Science Department, George Washington University, Washington, DC, United States of America, **2** ClustrX LLC, Washington, DC, United States of America, **3** Physics Department, George Washington University, Washington, DC, United States of America, **4** Institute for Data, Democracy & Politics, George Washington University, Washington, DC, United States of America, **5** Google LLC, Menlo Park, California, United States of America

* ylupu@gwu.edu

**Data Availability Statement:** We have created an online depository for our replication data. This is available at the Harvard Dataverse here: https://dataverse.harvard.edu/dataset.xhtml?persistentId=doi:10.7910/DVN/KKYTUN.

## Abstract

Online hate speech is a critical and worsening problem, with extremists using social media platforms to radicalize recruits and coordinate offline violent events. While much progress has been made in analyzing online hate speech, no study to date has classified multiple types of hate speech across both mainstream and fringe platforms. We conduct a supervised machine learning analysis of 7 types of online hate speech on 6 interconnected online platforms. We find that offline trigger events, such as protests and elections, are often followed by increases in types of online hate speech that bear seemingly little connection to the underlying event. This occurs on both mainstream and fringe platforms, despite moderation efforts, raising new research questions about the relationship between offline events and online speech, as well as implications for online content moderation.

## Introduction

Online hate and extremism are critical problems across the globe. Online, groups of like-minded, prejudiced individuals create communities on social media platforms, where they recruit new members and sometimes coordinate offline behavior. Users' exposure to hate speech can have important negative consequences [1]. At times, this *online* activity can fuel high-profile *offline* violence, such as the January 6th attack on the U.S. Capitol. In turn, highly charged *offline* trigger events tend to be followed by sharp increases in *online* hate speech [2, 3]. Such spikes in *online* hate speech have, further, been shown to predict similar spikes in *offline* violent hate crimes [4–9].

Insights from literatures on social movements can be applied to online extremist communities. These movements typically focus on the creation of a group identity, particularly framed to the exclusion of others [10, 11]. Hate speech plays a particularly important role by intensifying feelings of in-group cohesion and negative feelings toward the out-group [12]. By communicating with each other on social media, individuals affirm their own identities as well as those of the group, often by using specific language, symbols, and images [13, 14]. They use social media to communicate, mobilize, and coordinate collective action [15, 16]. These technologies may facilitate such collective action and make violence more likely [17, 18]. Elites,

**Funding:** YL and NFJ received funding from the Air Force Office of Scientific Research (FA9550-20-1-0382 and FA9550-20-1-0383) and National Science Foundation (SES-2030694). The funders had no role in study design, data collection and analysis, decision to publish, or preparation of the manuscript.

**Competing interests:** The authors have declared that no competing interests exist.

including the media, play a particularly important role in the process. Like other individuals, members of online hate communities take cues from media and other elites regarding both which issues and events to focus on and the opinions they might hold on those issues and events [19–24]. By making some offline events more salient, elites can thus focus the attention of online hate communities, so the relationship between offline trigger events and online hate speech is likely mediated in part by these actors. There is therefore a complex relationship between offline trigger events, elite/media cues and salience, online hate speech, and ensuing extremist violence that scholars across disciplines have sought to analyze and understand.

We study the connection between online hate speech and offline trigger events across the online ecology of moderated and unmoderated platforms. We focus on the following questions: How does online hate speech react to offline trigger events? Do these reactions differ across mainstream platforms that remove some hateful speech versus fringe platforms that are less likely to do so? Are offline events followed by online hate speech that is directly related to those events? Or, once hate speech has been kindled by an offline event, do members of online hate communities respond by using hate speech against seemingly unrelated targets?

Prior work has made significant progress on detecting and analyzing online hate speech. [Excellent surveys of this literature are available at [25, 26]]. With few exceptions [27, 28], existing work focuses on either classifying online hate speech as binary (i.e., hate or not hate) or classifying a small set of hate speech types [5, 29–31]. Most prior work tends to draw data from 1 or 2 social media platforms, usually mainstream and moderated platforms [31–36]. A recent survey found that Twitter was by far the most studied source in articles using automated detection of online hate speech [26]. Such approaches have two limitations we hope to mitigate. First, hate speech is likely to vary systematically between moderated and fringe platforms. Second, a more comprehensive classification of hate speech types is needed to better understand patterns of online hate speech following offline events.

We conduct a more comprehensive study of online hate speech by including a range of social media platforms and classifying a large set of hate speech types. We conduct a supervised machine learning analysis of 7 types of online hate speech using 59 million English-language posts drawn from 1,150 online hate communities on 6 social media platforms over the course of 19 months. This is a broad sample of online communities, particularly relative to prior work that tends to draw data from relatively few communities on moderated platforms. Our approach has several advantages. First, we include actively moderated mainstream platforms—Facebook, VKontakte, and Instagram—that have and enforce (to varying degrees) policies against hate speech, as well as the less-moderated platforms Gab, Telegram, and 4Chan. This allows us to capture online hate speech that would be filtered out of moderated platforms. Second, we collect data on posts on online hate communities across these platforms [37–39], allowing us to capture online hate speech where it is most likely to arise both before and after offline trigger events. Finally, by classifying a broad range of types of hate speech by their targets, we are able to analyze the extent to which offline events are followed by different types of online hate speech.

## Data

Our data collection effort begins by identifying online hate communities, which are online forums in which hate speech is most likely to be used. Many social media platforms offer individuals the means to join communities of like-minded users, such as fan pages on Facebook, channels on Telegram, and anonymous message boards on 4Chan. We identified candidate online communities by using the sampling procedure described in greater detail in the SI. Our team manually searched social media platforms for hate communities and search their content

to identify new communities. We included a community in our data as a hate community if 2 or more of the 20 most recent posts on the community included hate speech. For this step in the process, we defined hate speech as either (a) content that would fall under the provisions of the United States Code regarding hate crimes or hate speech according to Department of Justice guidelines; or (b) content that supports or promotes Fascist ideologies or regime types (i.e., extreme nationalism and/or racial identitarianism). The determination of whether an online community is a hate community was made manually using these criteria. We identified 1150 online hate communities that are included in this study.

Between June 1, 2019, and December 31, 2020, we captured the posts on these hate communities using a combination of automated and manual methods. We removed non-English-language posts by using Google's Compact Language Detector 2. Our data include the texts of 59 million posts, and do not include images, videos, and audio. The SI provides information on the distribution of posts and hate communities across the six platforms. We do not collect data on the geographic locations of users. A large share of them appears to be located in the United States based on their discussion of U.S. politics, but there are also frequent discussions of European issues and those of other English-speaking countries. We make no assumptions about the geographic locations of these users.

All data we collected were publicly posted on the Internet, and we had no contact with the users who posted the data. All data used in this study were collected and used in compliance with the terms of use of the online platforms. No personal identifying information was collected. No information about users was collected. User names were not collected. All the data are anonymous. We obtained advanced approval from the IRB.

Our study codes for 7 types of hate speech: speech targeting race, gender, religion, gender identity/sexual orientation (GI/SO), immigration, ethnicity/identitarian/nationalism (E/I/N), as well as anti-semitism. The SI provides additional information about how we define these types of hate speech. We chose these types of hate speech because, in our manual review of the hate communities, these types were both prevalent and distinguishable from each other (although they do co-occur).

## Methods

To construct our training data, four annotators were trained to manually code whether or not a post contained each of the 7 types of hate speech. We trained the enumerators to distinguish between mere discussion of an issue and hate speech, e.g., to distinguish between non-hateful discourse about race and racial hate speech. A post could, in principle, be coded as containing between 0 and 7 types of hate speech. In total, the training data include 31,323 manually coded posts; the SI provides additional information about the coding procedure and training data.

Using the algorithm developed by [40], we split these data into 26,354 posts used to train the machine and 4,969 used to test performance. We compared the performance of 8 model architectures with differing classification and text representation methods. Using each model architecture, we trained 7 distinct models, one for each type of hate speech, each of which separately learns from the training data. Each of these 7 models provides a binary coding indicating whether or not a given post contained the applicable type of hate speech or not. We then compared the performance of these 7 models across the 8 model architectures in order to choose the best-performing architecture. The best performing model architecture used a neural network classifier with BERT embeddings [41]. The accuracy ranged from 91.7% to 98.3%, depending on the type of hate speech. We validated the machine results using human annotators and found that the results were highly reliable. The results below are based on the 7 models using this model architecture. The SI provides additional information regarding the 8

model architectures we tested, a comparison of their performance, a detailed description of the performance of the model architecture we chose, and detail regarding the validation exercise.

## Results

We begin our analysis by focusing on the distribution of types of hate speech within the data. Panel A of Fig 1 shows how many posts contained each type of hate speech. By far the most common type of hate speech in the communities we tracked is racism, which appears in nearly 7 million posts. Most posts, despite originating from hate communities, did not contain any of the seven types of hate speech, as shown in Panel B. These communities often discuss unrelated issues, and some posts featured content other than text, which we did not analyze. About 20% of hate posts contained more than one type of hate speech. Panel C depicts how frequently each pair of hate speech types co-occurred within the same post. Most pairs are either weakly positively or weakly negatively correlated. The largest positive correlation is between race and E/I/N (0.2396), while the largest negative correlation is between gender and anti-semitism (-0.0226).

We next analyzed how hate speech changed over time. Fig 2 illustrates these changes in hate speech in these communities over time. The SI provides versions of these plots by platform. Panel A shows the temporal trend in the total count of daily posts across all 1150 communities. Activity in these communities differs systematically based on the day of the week, so here and elsewhere our results depict 7-day rolling averages. The total number of posts demonstrates an upward trend over the study period, increasing by about 67% from about 60,000 to 100,000 daily posts.

There are two especially notable spikes in total activity: in late May/early June 2020, and in early November 2020. They key offline events discussed in the posts during these periods were (1) the May 25 killing of George Floyd by Minneapolis police officers and the ensuing protests organized by the Black Lives Matter (BLM) movement; and (2) the November 2020 U.S. elections. Panel B shows that not only were there far more posts than usual during these two events, there were also increases in hate posts. Most importantly, these results indicate that during periods of intense and contentious offline activity, the activity in online hate communities increases dramatically. In part this be due to the heightened salience of these events more generally, i.e., these events were likely being discussed not only in these hate communities, but elsewhere online, leading to greater online activity in general, including hate speech activity.

Panel C, however, indicates that these two events had differing effects on the relative frequency of hate speech within these communities. During the early days of the BLM protests, the percentage of posts that contained hate speech increase, meaning that hate posts increased not only in absolute terms (as shown in Panel B) but also in relative terms compared to other posts. During the period around the U.S. elections, however, Panel C shows that the percentage of posts that contained hate speech decreased. There was an increase in hate posts (Panel B), but in relative terms their frequency decreased, i.e., posts increased overall at a greater rate than hate posts. Finally, Panel D shows that the mean number of hate speech types contained in hate posts has remained quite consistent over time. Two relatively small decreases occur during the events mentioned above.

We continue analyzing how online hate speech changed over time by focusing on each type of hate speech separately. Fig 3 shows the temporal trends in each type of hate speech. The SI provides versions of these plots by platform. Many events, occurring both offline and online, can affect online hate speech; here, we focus on the largest changes in the time-series during the study period. The largest spikes in online hate activity occurred around the following dates in 2020: (1) January 3 with respect to religion and antisemitism; (2) February 29 with respect

A. Total Posts Containing Each Type of Hate Speech

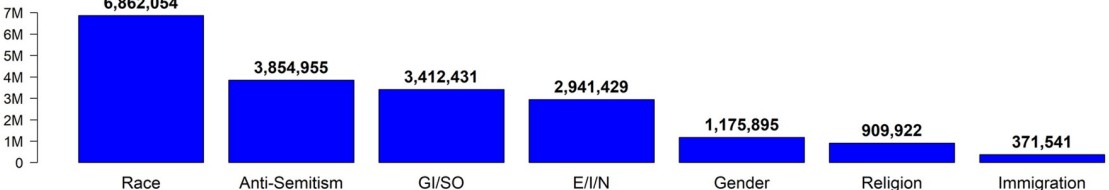

B. Total Posts Containing N Hate Speech Types

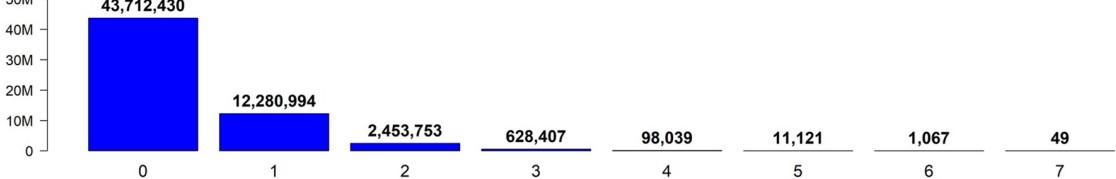

C. Co-Ocurrence of Hate Speech Types within Posts

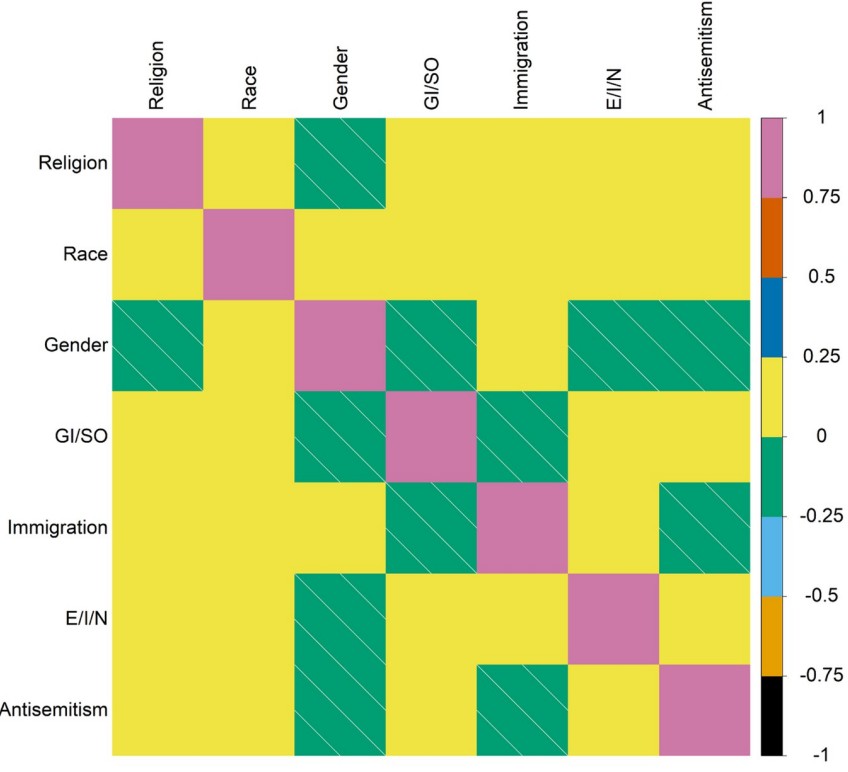

**Fig 1. Posts, hate posts, and types of hate speech.** Panel A shows how many posts contain each type of hate speech. Panel B shows how many posts contain different quantities of hate speech types. Panel C shows the pairwise correlations between hate speech types within posts. Diagonal lines indicate negative correlations.

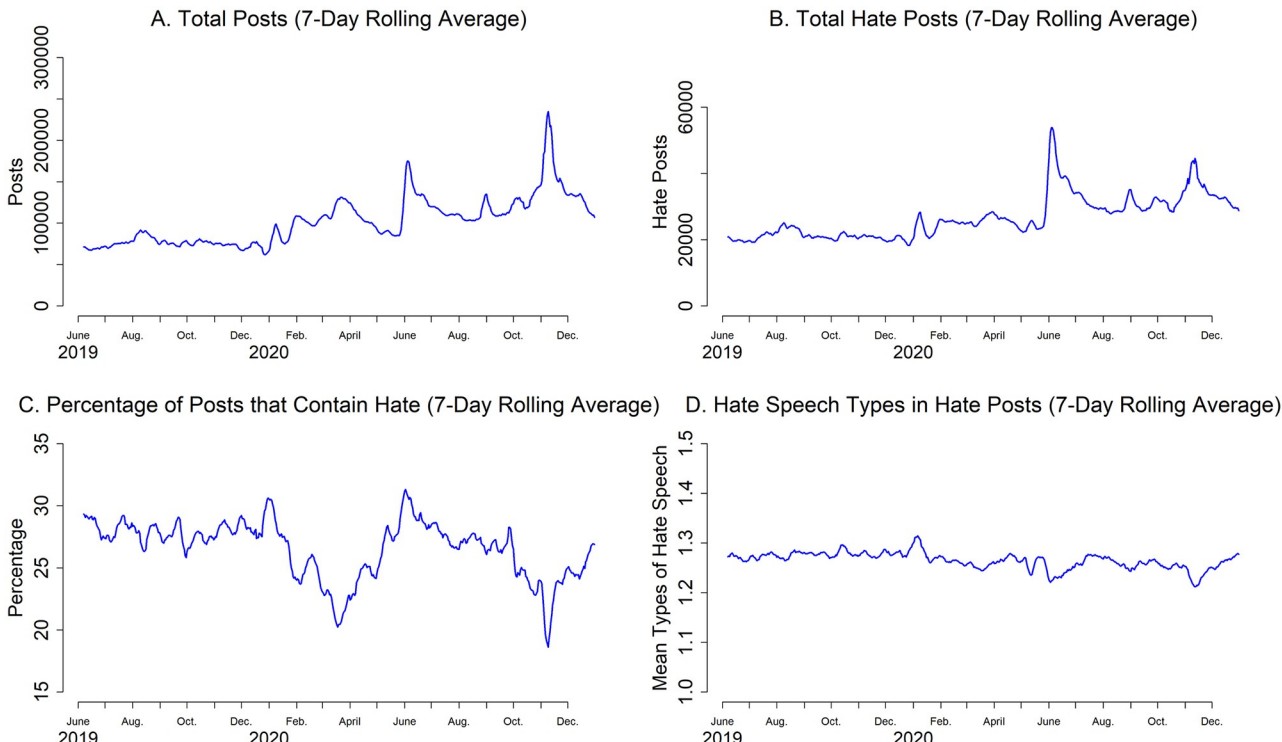

**Fig 2. Temporal trends in posts and hate posts.** Panel A shows the daily posts in the hate communities we tracked over time. Panel B shows the daily hate posts in these communities over time. Panel C shows the daily percentage of posts that were hate posts over time. Panel D shows the mean number of hate speech types in hate posts over time.

to immigration; (3) late May, with respect to race, gender, GI/SO, E/I/N, and antisemitism; and (4) early November, with respect to religion, gender, GI/SO, and immigration. Structural break analysis confirms that these changes in the data were not in line with prior patterns of hate speech (p<0.001) and are the largest breaks in the time-series.

We manually reviewed random samples of 500 hate posts from each of the periods surrounding these spikes and concluded they are associated with the following offline events: (1) the assassination of Iranian General Qasem Soleimani; (2) a crisis at the border of Turkey and Greece that gained publicity when a large number of mostly Syrian refugees sought entry to the European Union; (3) the George Floyd murder and ensuing protests; and (4) the November 2020 U.S. elections. Events 1 and 2 were less highly publicized than the latter two, and their effects on our data may be less intuitive. We therefore supplemented our manual analysis with an automated topic analysis to confirm the spikes in hate posts were related to those events. See the SI for details.

We focus our next analysis on these events. Each panel in Fig 4 shows the changes in online hate speech following one of these events. Because different types of hate speech are used at different rates, we normalize the level of each type of hate speech at 100 based on the seven-day rolling average prior to the event. Fig 4 therefore shows the relative changes in the frequency of each type of hate speech after each event. Following the Soleimani assassination, several types of hate speech increased significantly, with the largest increases in religion (90%) and anti-semitism (65%). Interestingly, the religious hate expressed in these posts was largely Islamophobic, whereas the rise in anti-semitism focused on speculation that Israel was behind the assassination. The change in hate speech activity during the Turkey-Greece border crisis is the

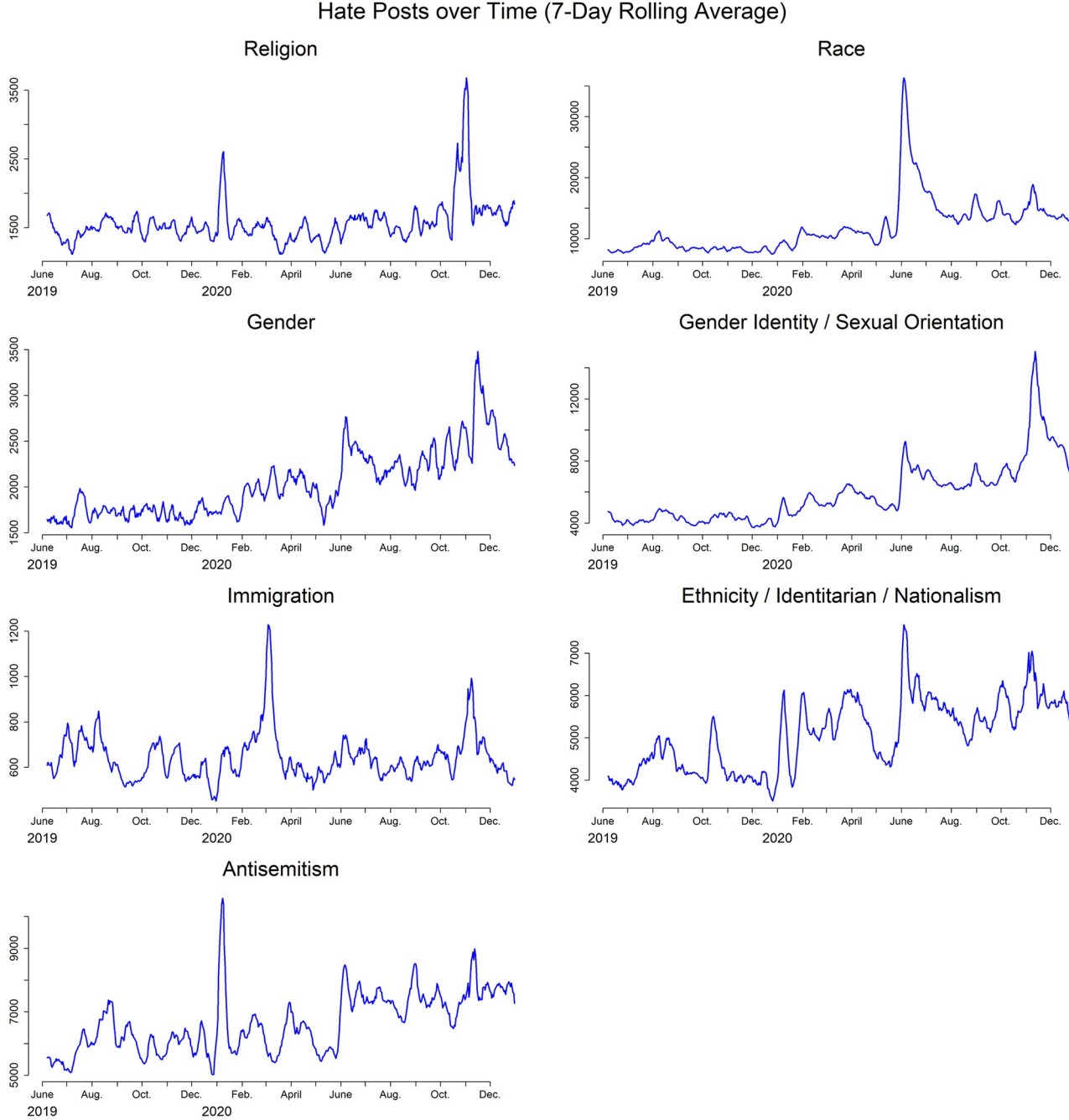

**Fig 3. Temporal trends in each type of hate speech.** Each panel shows the total daily posts that contain a given type of hate speech over time. Some types of hate speech appear much more often than others (Fig 1A), so the differences in y-axis scales (i.e., number of posts) should be noted.

narrowest of the events we studied: a large increase in anti-immigration hate speech, with relatively little change in other types.

The largest increases in hate activity during the study period followed the death of George Floyd and continued during the ensuing protests. The rate of racial hate speech increased by 250% by early June and, by the end of the year, remained about double what it had been prior

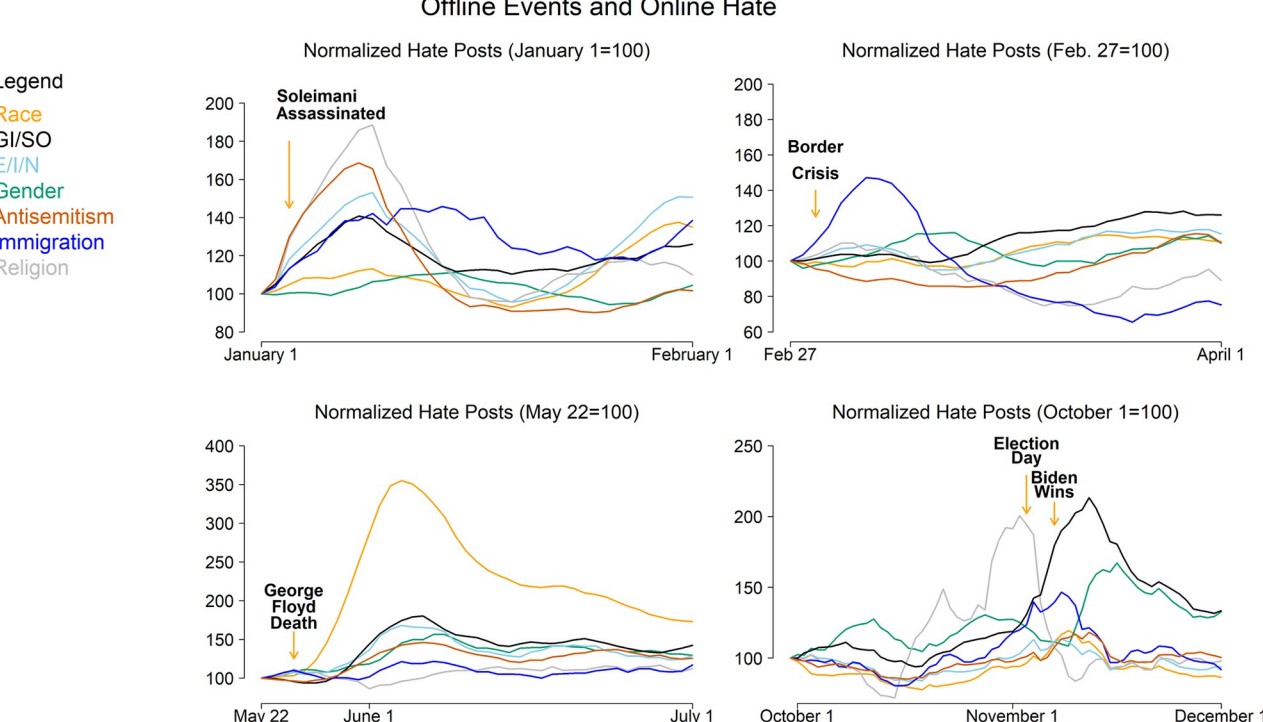

**Fig 4. Offline events and online hate.** Each panel shows the changes in the relative levels of each type of hate speech following a trigger event.

to this event. This is perhaps not surprising given the negative reaction to the protests in some segments of the public. What is surprising, however, is that the frequency of most other types of hate speech also increased dramatically at this time, especially GI/SO (75%), E/I/N (60%), gender (50%), and anti-semitism (40%). While Floyd's death and the protests were largely framed as being about racial issues, it is not intuitively clear what relationship these events have to issues of, for example, gender and sexual orientation.

The 2020 U.S. elections also present an interesting case. Unlike the other events we have examined, the election was scheduled, so online communities might react in anticipation of it. Thus, in the fourth subfigure of Fig 4, we normalize hate speech levels a month prior to the election rather than a few days prior. Immediately before the election there was large increase in religion hate, most of which is Islamophobic; this appears to be in response to several events in Europe and not related to the election. By Election Day (November 3) and continuing through the declaration of Joe Biden as winner by the Associated Press (November 7), there were large increases in GI/SO hate speech (over 100%) and immigration hate speech (50%); a wave of gender hate speech soon followed. In some ways, these waves of hate speech are surprising. Issues of gender identity and sexual orientation were not prominent during the campaign; the ensuing wave of associated online hate speech appears to be largely a case of users employing anti-LGBTQ slurs in a generalized manner to malign a wide range of political targets, such as candidates, parties, and voters. Likewise, immigration was a less prominent campaign issue than it had been in 2016, but after Biden's victory online hate speech increased due to speculation that he would moderate U.S. immigration policy. The ensuing wave of gender hate speech appears to be at least partly targeted toward Vice President Kamala Harris based on our manual review of the relevant posts. Interestingly, although racial issues were at the forefront during the campaign, online race hate speech changed relatively little before and

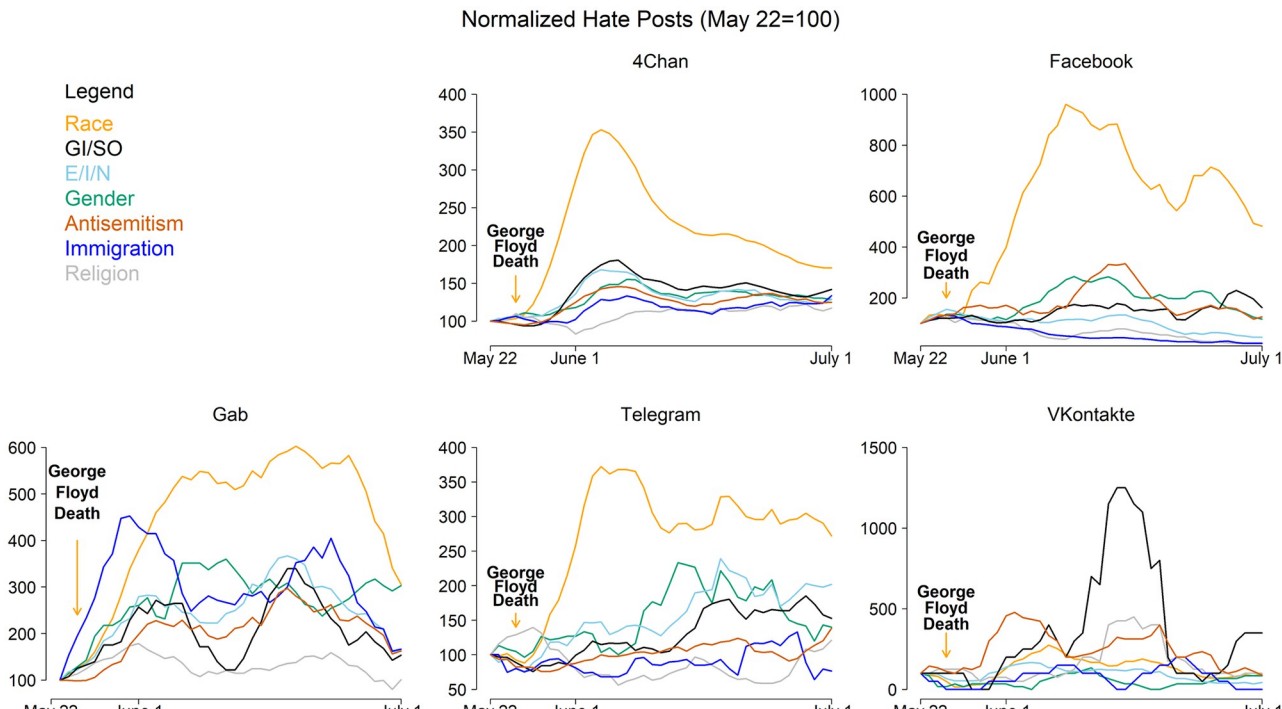

**Fig 5. Hate speech by platform after George Floyd's death.** Each panel shows the changes in the levels of each type of hate speech following the death of George Floyd on a given platform.

after the election. This may be because the rate of race hate speech, despite having dropped from its high in June, remained elevated throughout this period as compared to the period prior to George Floyd's death (see Fig 3).

Because hate activity increased to the greatest extent after Floyd's death, we focus in more detail on this event. Fig 5 shows results similar to those of Fig 4, but broken down by platform. Instagram is excluded from Fig 5 because there were not enough daily hate speech posts during this period to make meaningful inferences about changes in activity. The pattern on 4Chan largely follows the overall pattern across all the platforms because 4Chan makes up the majority of our data. More surprising is the relatively large increase in hate speech on Facebook. On Facebook, the daily average of posts containing race hate speech increased by a factor of 10 within days of Floyd's death and remained highly elevated (5x) a month later. Other types of hate speech, especially anti-semitism, gender, and GI/SO, also increased by large amounts on Facebook during these offline events. We observe similar patterns on Gab and Telegram: a very large increase in race hate speech along with smaller, but nonetheless notable, increases in several other types of hate that endure several weeks after the event. On VKontakte, however, there was relatively little reaction to these events, perhaps because users there are less attuned to U.S. politics. The increase in GI/SO hate activity on VKontakte in mid-June appears to be unrelated to these events.

## Discussion

Our findings have several implications for researchers and practitioners. The first is that online hate speech is a large and growing problem that affects both mainstream and fringe platforms. Over the study period, the daily volume of hate speech posts began at an average of 20,000,

peaked at around 50,000, and ended at around 25,000. Some prior work has found that the volume of extremist content and activity declined on YouTube, a moderated platform [42, 43]. Yet we find that, on the platforms we studied, including those that are moderated, hate speech increased despite elevated efforts to curtail it by moderated platforms, particularly during the second half of 2020. Although some platforms are increasingly committed to removing hate speech and other malicious content, hate content persists on such platforms (see, for example, Fig 9 in S1 File). Despite efforts by mainstream platforms to remove hate speech, it has grown in volume on fringe platforms such as 4Chan and Telegram (see Figs 8 and 10 in S1 File). Online extremists are known to migrate to less-moderated platforms after being removed from mainstream platforms [38, 44], which may be less susceptible to public and policymaker pressure.

It is therefore crucial to continue to analyze how mainstream platforms' content moderation affects the overall ecology of online hate speech. In particular, our findings raise several research questions. First, given the prior findings of decreases of extremist content on YouTube and our finding of increases in hate speech on other moderated platforms, we need a better of understanding of the conditions under which platforms can be more effective in moderating content. Key factors appear to include the type of content (e.g., hate speech as compared to other extremist content), the format of the content (e.g., text versus video), and the varying responsiveness of malicious content to offline events. In addition, given how quickly users are known to migrate across platforms, a key issue for future research is the overall impact of this migration on malicious content. When extremists and others who post malicious content are deplatformed from mainstream platforms, how many of them migrate to fringe platforms? Do only the most invested and extreme users do so, or do they bring a more casual following along with them? After such migration, does the nature of malicious content, such as hate speech, change? Does this content become more extreme, hateful, or otherwise malicious, or not?

Secondly, the spikes in online hate speech following offline events are not limited to fringe platforms. We know from prior work that online hate speech can react strongly following offline events [2], but our data allow us to compare the sizes of these increases across platforms and types of hate speech, which yields important new results. After the death of George Floyd, for example, the largest increase in online hate speech in absolute terms (i.e., the number of hate speech posts) was on 4Chan, but in *percentage terms* the largest increase was on Facebook (see Fig 5). This is a particularly striking result given Facebook's increased attention to content moderation. The implication of this for moderated platforms is that they should be especially watchful during such periods, but also that increases in hate speech on other, fringe platforms may be predictive of potentially similar increases on their own platforms, a point we hope to explore in future research.

Finally, and perhaps most importantly, we find that online hate speech reacts to differing offline events in different and sometimes counter-intuitive ways. We might conceptualize at least three types of offline events in terms of the patterns of hate speech that follow them. First would be offline events that could plausibly be followed by significant increases in hate speech (i.e., events involving frequent targets of hate speech), but that draw little reaction from online hate communities. For example, on March 13, 2020, Louisville, Kentucky, police fatally shot an African-American woman named Breonna Taylor in her home. The killing was followed by a relatively small discussion in the online hate communities we tracked–in contrast to the large reaction to the murder of George Floyd two months later. A second type of offline event is one that is followed by a discrete increase in online hate speech that appears to be directly related to the offline event, and then dissipates relatively quickly. The assassination of Suleimani and the Turkish-Greek border crisis appear to be events of this type. A third type of

offline event is followed by sharp increases in and broad cascades of online hate speech that build on, and potentially transform, the meaning and interpretation of the initial offline event and target groups in an extensive and seemingly indiscriminate manner. Events of this type can have longer-lasting effects on patterns and volume of online hate speech. In our study period, the murder of George Floyd and the U.S. Elections appear to be events of this type. It may be intuitive, for example, that online hate speech regarding race would increase during the BLM protests given the way those events were centered on the subject of race. Yet most other types of online hate speech also increased during that time, including, for example, GI/SO hate, which, on the surface, would appear to have an indirect connection to issues of police brutality and race.

## Conclusions

What factors explain the different relationships between individual offline events and online hate speech? A detailed analysis of this question is beyond the scope of this paper, but media and elite attention likely play an important role, as discussed in the introduction. Both the volume and variety of online reactions to offline events depend, in part, on the salience of those events in other media. For example, the killing of Taylor received little attention in the U.S. national media in the weeks that followed it (perhaps because it occurred during the time when many U.S. jurisdictions were announcing the first lockdowns of the COVID-19 pandemic), and became significantly more salient during the protests that followed the Floyd murder. The Soleimani assassination and EU border crisis received media attention, but for a shorter time span than the BLM protests and U.S. election. These observations suggest that media attention may be one of the key factors that explains why the patterns of online hate speech following individual offline events differ so sharply.

Yet this factor likely does not explain the range of our findings. For example, if online racist hate speech was, in part, activated by media attention to the BLM protests that made racial identity a more salient issue, why did other forms of hate speech also increase dramatically at that time? Why does this broad activation of online hate speech occur following some offline events but not others? We hope future work will investigate these questions. A particularly important question for future work is to investigate which *types of offline events* are more likely to be followed by broad and seemingly indiscriminate cascades of online hate speech. Likewise, our results also raise the question of which *types of online hate speech*, once triggered by such events, are more likely to be followed by such a cascade into other types of hate speech.

Our study faces two important limitations. It is not possible to identify the entire population of online hate communities and sample from it. The online ecology of social media platforms is vast and constantly changing. Tracking and capturing data on these platforms is also limited by terms-of-service and privacy concerns. It is therefore not possible to capture the entire ecology of online speech nor to confidently claim that any sample of such speech is representative of the ecology. Our list of online hate communities is nonetheless the most comprehensive such list we are aware of. After several months of search by several members of our team, we are not aware of any other communities on the 6 platforms we studied that meet our definition of hate communities. We have captured many of the most prominent and active online communities on which hate speech is regularly used. Our findings and conclusions are limited to the public communities we identified. We can infer from our results the relationship between offline events and the use of hate speech on our sample of prominent online hate communities, but we cannot infer from our results, nor from those of prior work, whether or not this relationship holds across the online ecology.

A second limitation of our work is that our analysis does not allow for causal inference between observed offline and online phenomena. We reviewed the posts associated with the spikes in online hate speech discussed above and concluded they are substantively related to the trigger events we described. Following the death of George Floyd protests, for example, the large increase in posts that contained race hate speech (see Fig 4) included many posts that referenced Floyd, the circumstances surrounding his death, and the ensuing protests, leading us to conclude the increase in online hate speech was related to these offline events.

We conclude by noting that much work remains to be done to better understand the relationship between offline events and online hate speech. We have conducted a wide-ranging analysis of online hate speech–across types of hate speech, many online communities, and a diverse set of platforms. We have provided evidence that suggests complex relationships between offline events and online hate, between different types of online hate, and between moderated and fringe platforms. These results raise important questions for scholars across multiple disciplines and suggest several strategies for mitigating online hate speech. First, content moderators should be increasingly watchful of offline events, and efforts may need to be augmented following highly salient events. Second, the effects of an offline event on online hate speech may be difficult to predict, and may quickly evolve and broaden, which further suggests a broad and rapid mitigation effort in response. Because online hate speech levels are predictive of offline hate crimes [4–9], the potential exists for an offline trigger event to be followed by a broad cascade of online hate speech that, in turn, precedes hate crimes of a nature unrelated to the initial event. Third, because the effects of an offline event on different types of platforms are often similar, content moderators on mainstream platform could advance their efforts by systematically observing ongoing developments on other, fringe platforms.

## Supporting information

**S1 File. Supplementary material to the manuscript.**
(PDF)

## Author Contributions

**Conceptualization:** Yonatan Lupu, Nicolas Velásquez, Beth Goldberg.

**Data curation:** Richard Sear, Rhys Leahy, Nicholas Johnson Restrepo.

**Formal analysis:** Yonatan Lupu.

**Funding acquisition:** Yonatan Lupu, Neil F. Johnson.

**Investigation:** Yonatan Lupu, Nicholas Johnson Restrepo.

**Methodology:** Yonatan Lupu, Richard Sear, Nicolas Velásquez.

**Project administration:** Yonatan Lupu, Beth Goldberg.

**Resources:** Yonatan Lupu.

**Software:** Richard Sear, Nicolas Velásquez.

**Supervision:** Yonatan Lupu, Beth Goldberg, Neil F. Johnson.

**Validation:** Richard Sear, Rhys Leahy.

**Visualization:** Yonatan Lupu.

**Writing – original draft:** Yonatan Lupu, Rhys Leahy.

**Writing – review & editing:** Yonatan Lupu.

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
