## [Decision Letter · Decision Letter 0]

1 May 2022

PONE-D-22-01701Offline Events and Online Hate PLOS ONE

Dear Dr. Lupu,

Thank you for submitting your manuscript to PLOS ONE. After careful consideration, we feel that it has merit but does not fully meet PLOS ONE’s publication criteria as it currently stands. Therefore, we invite you to submit a revised version of the manuscript that addresses the points raised during the review process. Specifically, pay particular attention to the doubts that all reviewers raised about the methodology and to the need to better frame and motivate the manuscript. The former point includes providing thorough information on how the data was chosen, collected, and labeled, and providing adequate evidence of the classifiers' accuracy.

We look forward to receiving your revised manuscript.

Kind regards,

Stefano Cresci

Academic Editor

PLOS ONE

Journal Requirements:

2. Please provide additional details regarding participant consent. In the Methods section, please ensure that you have specified (1) whether consent was informed and (2) what type you obtained (for instance, written or verbal). If the need for consent was waived by the ethics committee, please include this information.

*In line with our policy regarding third-party data use (https://journals.plos.org/plosone/s/submission-guidelines#loc-personal-data-from-third-party-sources), please provide a copy of the protocol that your IRB reviewed and approved, and which specifies what types of data you collected and how data were anonymized. Please submit this as an ‘Other’ file, as this will be only for our records and will not be published."

- please also ping me and the Data team at PRTC to discuss the data sharing/availability options for this ms

3. We note that you have stated that you will provide repository information for your data at acceptance. Should your manuscript be accepted for publication, we will hold it until you provide the relevant accession numbers or DOIs necessary to access your data. If you wish to make changes to your Data Availability statement, please describe these changes in your cover letter and we will update your Data Availability statement to reflect the information you provide

Additional Editor Comments (if provided):

Thank you for giving us the chance to evaluate this manuscript. All reviewers found merit in this submission. However, they all also highlighted several major shortcomings that need to be addressed. Among them, pay particular attention to the doubts that all reviewers raised about the methodology and to the need to better frame and motivate the manuscript. The former point includes providing thorough information on how the data was chosen, collected, and labeled, and providing adequate evidence of the classifiers' accuracy. Good luck for the revision.

Reviewers' comments:

Reviewer's Responses to Questions

**Comments to the Author**

1. Is the manuscript technically sound, and do the data support the conclusions?

Reviewer #1: Partly

Reviewer #2: Yes

Reviewer #3: Yes

2. Has the statistical analysis been performed appropriately and rigorously? 

Reviewer #1: No

Reviewer #2: Yes

Reviewer #3: Yes

3. Have the authors made all data underlying the findings in their manuscript fully available?

Reviewer #1: Yes

Reviewer #2: Yes

Reviewer #3: No

4. Is the manuscript presented in an intelligible fashion and written in standard English?

Reviewer #1: Yes

Reviewer #2: Yes

Reviewer #3: Yes

5. Review Comments to the Author

Reviewer #1: This manuscript addresses an important topic: how hate speech responds to offline events. It also engages in measurement of cross-platform hate speech directed at multiple targets. While I am generally very supportive of such efforts to systematically measure and track online speech, I have several concerns, which I detail below.

First, there are several aspects of the authors’ sampling approach that they should do more to explain. Why do the authors look at only one community on 4chan? This seems particularly odd given that the vast majority of the authors’ training data comes from 4chan. How might this bias the authors’ ability to detect hate speech across platforms? The majority of their training data also focuses on racist hate speech in particular, but is this an artifact of using almost entirely data from a single 4chan community? This likely limits the authors ability to detect other hate speech types in their data, particularly across platforms. More generally, what is the advantage in focusing on hate groups rather than representative samples of posts across each platform (see Siegel et al. 2021). The authors need to do more to justify this approach.

Second, given how rare many types of hate speech are in the authors’ data, it seems the classifier performance metrics (often in the high 90s) are likely inflated. I would like to see balanced accuracy metrics to better understand whether high classifier performance is simply an artifact of having so many 0s in the data. The authors should also conduct systematic validation exercises, taking random samples of their data and using human coding to determine how accurate their classifier performance is in practice.

Third, the authors focus on measuring changes in volume over time makes it very difficult to interpret their results. In the aftermath of an event that makes a particular group identity more salient (eg George Floyd protests), we should expect to see large increases in discussions of that group (eg African Americans) online. This would lead to an increase in hate speech volume simply because the group is discussed more often and a certain percentage of posts about that group will always include hate speech online. Indeed, when the authors do look at their results in percentage terms, they yield very different findings. The authors should do more to describe changes in absolute vs relative terms in their data and interpret any results referencing absolute changes with caution.

Reviewer #2: This paper advances the state-of-the-art on Hate-Speech characterization by analysing seven types of online hate speech on a set of six social media which includes both moderated and unmoderated platforms; moreover, it investigates the connection between offline and online hate speech.

The research questions are very interesting, and the findings are effectively supported by plots and the written discussion. Also, the data collection is very accurate. Though I have got some stylistic suggestions to give and doubts about the classification methods to solve, I believe that this work is extremely valuable.

Stylistic suggestions:

1) As far as it concerns the paper’s structure, the ‘Introduction’ title should be written (maybe it’s an oversight), and the Conclusion section should be added; also, it would be great to separate the data collection and the methods, making it clear that the methods are about the multi-label classification of posts.

2) Move the discourse of lines [67-69] right before “Second, by classifying …” at line 65, because you’re talking about the six platforms again.

3) You could start the ‘Classification Method’ section including lines [106-121]. Choose an appropriate section title.

4) Before stating which is the best classifier model at line 115, explain in more details the goal of that model:

4.1) Generally speaking, you built 6 models whose goal is classifying the 50k posts according to the seven hate-speech types, or assigning the ‘no hate speech is present’ label. You should explicit this goal.

4.2) Clearly explain how the model is used to derive the 7 machine learning algorithms; additionally, it’s not quite understandable what are those 7 machine learning algorithms.

4.3) Finally, if for each hate-speech type you used a binary classifier telling whether a post does or does not contain a given type of hate speech, how do you label a post as ‘hate-speech not detected’?

5) Figures:

5.1) Provide color-blind friendly figures. For example, use different markers, line sizes, grey shades.

5.2) Provide higher quality images following PLOS ONE’s guidelines.

5.3) Fig 1, panel B: it needs a better title, like “Total posts containing N hate speech typologies”; add the x-axis label, something like “N Hate Speech Typologies” or “How Many Hate Speech Typologies”

5.4) Fig 1, panel C: you cannot clearly detect the ‘-‘ sign before the ‘0.1’ at the end of the scale. Besides, the negative values (negative correlation) should be well separated from the positive ones, otherwise the correlation between Gender and Religion (to cite an example) is doubtful. Moreover, what does the dots’ size mean?

6) Line 212: specify that you are talking about the fourth subfigure in Fig. 4

7) Supporting Information document:

7.1) Fig 1, panel B: same suggestions as for the main paper

7.2) Maybe the ‘Machine Learning Methods and Model Selection’ section could benefit from a figure showing how the generic model is used to classify each post with hate-speech types. I did not understand how the ‘weights’ are assigned to perform binary classification (at the end of the first paragraph of this section)

7.3) The accuracy should be used only when the dataset is balanced, hence Fig. 2 may show the F1 score instead of the accuracy.

Classification method doubts:

1) if for each hate-speech type you used a binary classifier telling whether a post does or does not contain a given type of hate-speech, how do you label a post as ‘hate-speech not detected’? From what I understand, a post is labelled as ‘hate-speech not detected’ if for all seven binary classifiers that post is classified as ‘hate-speech not detected’, right?

2) Inside the Supporting Information: “[…] whose output is used for 7 separate densely-connected layers […]”, there are 7 separated layers because each one is used to detect a hate speech type, right?

Reviewer #3: The paper "Offline Events and Online Hate" is a well written manuscript that deals with a highly significant issue in social science research by using data science techniques. I would stress two main merits of this research: (i) the chosen approach that looks into the relationship between "real" events and the online discursive world, which is one of the main challenges of social science research for the next years, given the fact that these two stages are fully interconnected but we are not sure yet how this imbrication works; (ii) the huge collected data with an unprecedented survey of online hate communities across different platforms (and not only Twitter!) and the specificity of hate speech (divided into 7 categories).

However, the manuscript has some important shortfalls that the authors should address:

1. The title is ambitious and the empirical findings do not provide enough evidence to explain the relationship between offline events and online hate (the authors themselves write this is beyond the the scope of the paper). This initial consideration about the title also affects the general narrative of the manuscript.

2. The rationale of the paper is not only short but weak. There is not mention to any theoretical background that help to understand the relationship between offline events and online discourses, which in turn would provide a better justification of the RQ's. Which social science theoretical approaches can support the ideas or hypotheses included in this research? Even with a data-driven approach this paper should reflect on this important matter: How can we theoretically model the connections between offline events - media - hate speech - hate crimes? In fact, the authors stressed the importance of what they call "media attention" to understand the connection between the first three elements, but there not a single mention to well-grounded theoretical approaches that already explain part of this process, such as mediatization or, more specifically, the Agenda-Setting hypothesis. If offline events affect the amount of hate speech through the media, some attention should be payed to explain this relationship. Interestingly, the seminal research also suggest that hate speech derives in more hate crimes, which is a offline event again but of a different nature. Probably, including the theoretical understanding of this (recursive) pipeline would be of significant help for the scientific community.

3. The manual classification for the training corpus (made by 4 coders) should be better explained (in SI or in the paper). First, which kind of training received these coders?, were they supervised (i.e. by choosing and double-checking a random sample of the messages they coded)?; Second, and more important, what is the reliability of the manual annotation?, is there an inter-coder agreement test that guarantees the reliability of the coding process? Were some messages coded simultaneously by 2 or more coders?, and if it was the case, what happened to those messages in which there was not agreement? This may seem a small detail in the method, but I consider that the first step to obtain trust in the ML models is to have a really good training corpus and to be sure the the manual classification was of high quality.

4. Regarding the "manual review" of the hate posts, the authors should better explain how it was done (qualitatively, I guess) (and how many of them were reviewed before going to topic modeling) and specially how they linked the topics to the offline events. Did they build a list of offline events? Or how they decided to mention the set offline events included in the manuscript and not some others. Could they bring some agenda-setting studies to get the events with more media attention during the studied period? Moreover, when linking the spikes of hate speech to the events using the main topic detected with LDA, I wonder which other topics were found (I didn't see it in the SI) and if they could be meaningful in some way.

5. When choosing the ML models, I wonder why specific NN were not tested, even when they are state-of-the-art in supervised text classification. Specifically, I am referring to RNN and CNN, which had been used in many previous studies of hate speech detection.

6. Some minors details should be addressed such as the correct references to figures in line 266 (p. 17)

6. PLOS authors have the option to publish the peer review history of their article (what does this mean?). If published, this will include your full peer review and any attached files.

Reviewer #1: No

Reviewer #2: No

Reviewer #3: No

---

## [Author Response · Author response to Decision Letter 0]

9 Nov 2022

Please see the response memo at the end of this pdf for a detailed response to the editor and reviewer comments. Thank you.

---

## [Decision Letter · Decision Letter 1]

18 Nov 2022

Offline Events and Online Hate

PONE-D-22-01701R1

Dear Dr. Lupu,

We’re pleased to inform you that your manuscript has been judged scientifically suitable for publication and will be formally accepted for publication once it meets all outstanding technical requirements.

Kind regards,

Hossein Kermani

Academic Editor

PLOS ONE

Additional Editor Comments (optional):

Congratulations! 

Reviewers' comments:

Reviewer's Responses to Questions

**Comments to the Author**

1. If the authors have adequately addressed your comments raised in a previous round of review and you feel that this manuscript is now acceptable for publication, you may indicate that here to bypass the “Comments to the Author” section, enter your conflict of interest statement in the “Confidential to Editor” section, and submit your "Accept" recommendation.

Reviewer #2: All comments have been addressed

Reviewer #3: All comments have been addressed

2. Is the manuscript technically sound, and do the data support the conclusions?

Reviewer #2: Yes

Reviewer #3: (No Response)

3. Has the statistical analysis been performed appropriately and rigorously? 

Reviewer #2: Yes

Reviewer #3: (No Response)

4. Have the authors made all data underlying the findings in their manuscript fully available?

Reviewer #2: No

Reviewer #3: (No Response)

5. Is the manuscript presented in an intelligible fashion and written in standard English?

Reviewer #2: Yes

Reviewer #3: (No Response)

6. Review Comments to the Author

Reviewer #2: The authors addressed all comments and requests by better structuring the manuscript and detailing the data gathering and methodology in an exhaustive way. The analysis has been performed appropriately, leading to interesting results and discussions.

As for the data availability, the authors decided to share them only after the acceptance of the manuscript.

Reviewer #3: All my comments have been addressed. The paper is now ready for publication. The manuscript constitutes a contribution to the field.

7. PLOS authors have the option to publish the peer review history of their article (what does this mean?). If published, this will include your full peer review and any attached files.

Reviewer #2: No

Reviewer #3: **Yes: **Carlos Arcila Calderón

---

## [Editor Report · Acceptance letter]

23 Dec 2022

PONE-D-22-01701R1 

Offline Events and Online Hate 

Dear Dr. Lupu:

I'm pleased to inform you that your manuscript has been deemed suitable for publication in PLOS ONE. Congratulations! Your manuscript is now with our production department. 

Kind regards, 

on behalf of

Dr. Hossein Kermani 

Academic Editor

PLOS ONE